# AugMax: Adversarial Composition of Random Augmentations for Robust Training

**Haotao Wang**[1]*, **Chaowei Xiao**[2,3], **Jean Kossaifi**[2], **Zhiding Yu**[2],
**Anima Anandkumar**[2,4], and **Zhangyang Wang**[1]

[1]Department of Electrical and Computer Engineering, University of Texas at Austin
[2]NVIDIA  [3]Arizona State University  [4]California Institute of Technology
[1]*{htwang, atlaswang}@utexas.edu*
[2]*{chaoweix, jkossaifi, zhidingy, aanandkumar}@nvidia.com*

## Abstract

Data augmentation is a simple yet effective way to improve the robustness of deep neural networks (DNNs). Diversity and hardness are two complementary dimensions of data augmentation to achieve robustness. For example, AugMix explores random compositions of a diverse set of augmentations to enhance broader coverage, while adversarial training generates adversarially hard samples to spot the weakness. Motivated by this, we propose a data augmentation framework, termed AugMax, to unify the two aspects of diversity and hardness. AugMax first randomly samples multiple augmentation operators and then learns an adversarial mixture of the selected operators. Being a stronger form of data augmentation, AugMax leads to a significantly augmented input distribution which makes model training more challenging. To solve this problem, we further design a disentangled normalization module, termed DuBIN (Dual-Batch-and-Instance Normalization), that disentangles the instance-wise feature heterogeneity arising from AugMax. Experiments show that AugMax-DuBIN leads to significantly improved out-of-distribution robustness, outperforming prior arts by 3.03%, 3.49%, 1.82% and 0.71% on CIFAR10-C, CIFAR100-C, Tiny ImageNet-C and ImageNet-C. Codes and pre-trained models are available: `https://github.com/VITA-Group/AugMax`.

## 1 Introduction

Out-of-distribution (OOD) samples present a challenge when deploying AI models in the real world. Examples include natural corruptions (*e.g.*, due to camera blurs or noise, snow, rain, or fog in image data), sensory perturbations (*e.g.*, sensor transient error, electromagnetic interference) and domain shifts (*e.g.*, summer $\rightarrow$ winter). However, deep networks are often trained on limited amounts of data which may not cover sufficient scenarios. As a result, they are vulnerable to unforeseen distributional changes despite achieving high performance on standard benchmarks [1–3]. This jeopardizes their trustworthiness as well as safe deployment in real-world environments. Thus it is critical to develop techniques that improve robustness even when training with relatively clean datasets.

Several techniques have been proposed to consolidate the robustness against unforeseen corruptions, including robust data augmentation [5, 3, 8, 6], Lipschitz continuity [9–11], stability training [12], pre-training [13–16], and robust network structures [17–19], to name a few. Among these techniques, data augmentation is of particular interest due to its empirical effectiveness, ease of implementation, low computational overhead, and plug-and-play nature.

---

*Work partially done during an internship at NVIDIA.

35th Conference on Neural Information Processing Systems (NeurIPS 2021).

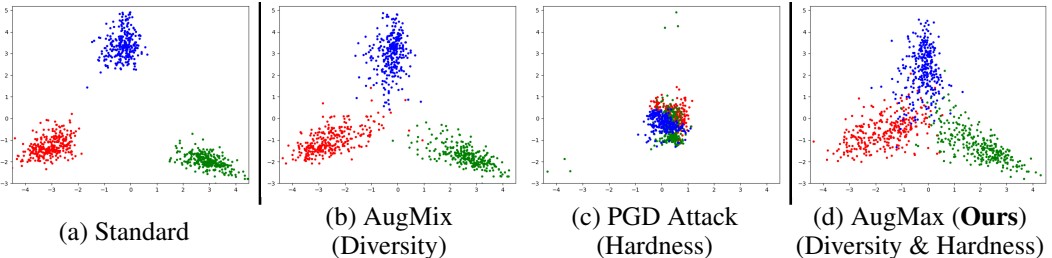

|(a) Standard | (b) AugMix (Diversity) | (c) PGD Attack (Hardness) | (d) AugMax (**Ours**) (Diversity & Hardness)|

Figure 1: **The effects of diversity and hardness in data augmentations.** We visualize features of augmented images fed to the network during training. The features are from the penultimate layer of a ResNeXt29 trained on CIFAR10 training data using only standard data augmentation (random flipping and translation) following [4]. For visualization, we randomly selected 300 fixed images from 3 fixed classes (denoted by different colors) to which we apply different augmentation methods: (a) standard augmentation (random flipping and translation); (b) AugMix [5]; (c) PGD Attack [6, 7]; (d) AugMax (ours). In order to achieve good model robustness, the augmented training data should both be diverse and also contain enough hard cases. As can be seen, PGD Attack generates hard cases (which the network cannot separate) but are not diverse enough (they are all clustered together) while AugMix creates diverse but not hard samples (they are well separated). By contrast, our approach (AugMax) achieves a unification between hard and diverse samples.

There are mainly *two categories* of data augmentation approaches:

**The first category** aims to increase *diversity* of the training data by composing multiple random transformations [20–22, 5, 3]. While standard data augmentation methods (*e.g.*, random flipping and translation) lead to poor robustness [5], more aggressive combinations of multiple augmentations have shown promise. One such successful example is AugMix [5], which stochastically samples from diverse augmentation operations and randomly mixing them to produce highly diverse augmented images. AugMix can increase the sample diversity coverage compared to standard augmentations, as shown in Figure 1 (a) and (b).

**The second category** aims to boost the *hardness* of the training data by sampling from the worst-case augmentations that tries to fool the model into misclassifying the samples. A common technique to achieve this goal is adversarial perturbation [6, 23, 8]. Training over such worst-case samples allows a model to actively fix its generalization weaknesses [24] and empirically improves its robustness against corruptions [25, 26, 7]. Due to the extra complexity to generate adversarial perturbations, the improved robustness usually comes at the cost of largely increased training time compared with non-adversarial methods [6, 27]. An example falling into this category is the PGD attack [6], as is shown in Figure 1 (c).

Previous work has focused on leveraging one of these category to improve robustness. In this paper, we *unify* both approaches in a single framework. In particular, we show that diversity and hardness are *complementary* and that a unification between the two is necessary to achieve robustness. We propose a new strategy to achieve this unification and successfully increase robustness.

**Summary of contributions**:

- We propose **AugMax**, a novel augmentation framework which achieves robustness through a unification between diversity and hardness, by searching for the worst-case mixing strategy.

- Being a stronger form of data augmentation, AugMax leads to a significantly augmented and more heterogeneous input distribution, which also makes model training more challenging. To solve this problem, we design a new normalization strategy, termed **DuBIN**, to disentangle the instance-wise feature heterogeneity of AugMax samples.

- We show that combination of AugMax and DuBIN (**AugMax-DuBIN**) achieves state-of-the-art robustness against corruptions and improves robustness against other common distribution shifts.

AugMax achieves a good unification between sample diversity and hard corner-case generation during data augmentation. AugMax is built on top of the AugMix framework [5] which mixes multiple data augmentation operators in a multi-branch and layered pipeline. However, different from AugMix where augmentation operators and mixing weights are both randomly sampled, operators are first

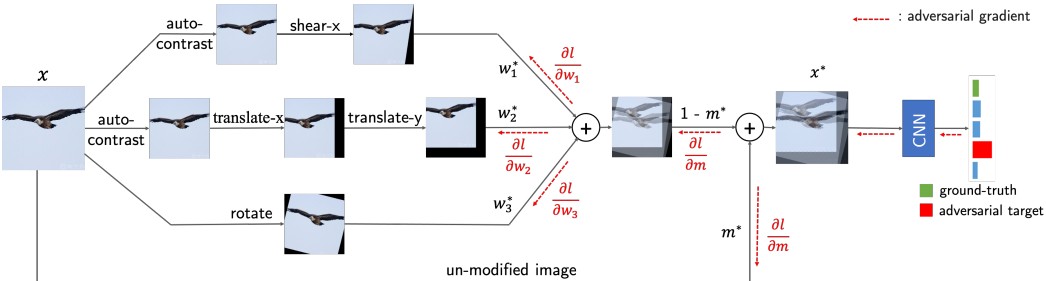

Figure 2: AugMax overview. Black and red arrows represent forward paths and back-propagation paths to generate AugMax images, respectively. In contrast to AugMax, where the mixing parameters $m$ and $w$ are adversarially learned, AugMix randomly samples $m$ and $w$ from predefined distributions (and thus no backpropergation on $m$ and $w$).

randomly sampled, followed by adversarially trained mixture of the selected operators in Augmax. This simple change from AugMix to AugMax results in considerable difference in their feature distributions. From the visualizations (in Figure 1), AugMax generates more adversarially "hard samples", while still keeping a good amount of diversity compared to AugMix and PGD Attack. Searching for adversarial mixing strategies in AugMax is slightly more expensive since it involves adversarial mixing. To make this efficient, we adopt an efficient adversarial training strategy [28]. As a result, AugMax adds only a reasonable amount of extra complexity compared to non-adversarial training methods, while improving robustness significantly. For example, AugMax training time is only $\sim 1.5$ times that of AugMix on ImageNet (see Table 7). Moreover, Augmax is significantly more efficient than traditional adversarial training which has $\sim 10$ times training time overhead compared with AugMix.

Being a stronger form of data augmentation with adversarial sample generation, AugMax leads to a significantly augmented input distribution which makes model training more challenging. This naturally motivates us to propose a novel and finer-grained normalization scheme termed Dual-Batch-and-Instance Normalization (DuBIN). As illustrated in Figure 3, DuBIN adds an additional instance normalization (IN) in parallel to the traditional Dual Batch Normalization (DuBN) [7, 29] used in adversarial training, in order to better model and disentangle the instance-wise feature heterogeneity arising from AugMax. We show that adding the instance normalization is an important knob to promote the capability of AugMax in boosting model robustness.

Our framework, AugMax-DuBIN, is illustrated in Figure 2. AugMax-DuBIN trains on clean images and achieves state-of-the-art robustness on natural corruption benchmarks [1], and also improves model robustness against other common distribution shifts [30, 2]. In particular, our method surpasses state-of-the-art method on CIFAR10-C, CIFAR100-C, Tiny ImageNet-C and ImageNet-C by 3.03%, 3.49%, 1.82% and 0.71% respectively.

## 2 Related Work

### 2.1 Robustness to Distributional Shifts

Hendrycks *et al.* [1] pioneer the study of prediction errors exhibited by machine learning models on unseen natural corruptions and perturbations. They introduce two variants of the original ImageNet validation set to evaluate the model's robustness: the ImageNet-C for input corruption robustness, and the ImageNet-P for input perturbation robustness. The former applies 15 diverse corruptions drawn from four main categories (noise, blur, weather, and digital) on ImageNet validation images. The latter contains perturbation sequences generated from each image. Recht *et al.* [30, 2] collect CIFAR10.1 and ImageNet-V2 as a reproduction of the original CIFAR10 [31] and ImageNet [32] test sets, and find that a minute natural distribution shift caused by the minutiae in dataset collection process leads to a large drop in accuracy for a broad range of image classifiers. Geirhos *et al.* [33] observe that deep models trained on ImageNet are biased towards textures, and the model robustness can be improved by emphasizing more on global shape features [33–36]. Wang *et al.* [37] find that the benchmark test performance does not always translate to the real-world generalizability,

and propose an efficient method to troubleshoot trained models using real-world unlabeled images. Hendrycks *et al.* [38] collect ImageNet-A with the new concept of natural adversarial examples: unmodified real-world images that falsify common machine learning models. Other representative benchmark efforts include [39–42, 3], all demonstrating the brittleness of deep models under various distribution shifts. Readers of interest are referred to a recent survey [43].

## 2.2 Data Augmentation: Random and Adversarial

Data augmentations have been widely used to increase training set diversity and improve model generalization ability [44, 21, 33, 22]. Among those methods, AugMix [5] achieves the outstanding robustness against natural corruptions by randomly mixing multiple diverse augmentations. It establishes high performance bars on many natural corruption benchmarks including CIFAR10-C, CIFAR100-C and ImageNet-C [1]. DeepAugment [3] feeds clean images to image-to-image models with randomly perturbed model weights to generate visually diverse augmented images. Combined with AugMix, it achieves state-of-the-art robustness on ImageNet-C [1]. Lately, a concurrent work MaxUp [45] generates hard training samples by selecting the worst-case weights in the MixUp [20] pipeline, to improve both model generalization ability and adversarial robustness. Our work shared a similar mindset to MaxUp, but we build on the more sophisticated AugMix pipeline and handle the resulting higher feature heterogeneity using a new normalization module.

Besides, Adversarial training (AT) [6, 46–52] utilizes adversarial samples as a special data augmentation method, and has also shown promise in improving model robustness against natural corruptions [23] and domain gaps [25], usually at the cost of increasing training time [27] and degrading the standard accuracy on clean images [53]. Volpi *et al.* [25] and Zhao *et al.* [26] augment the training set with samples from a fictitious adversarial domain to improve domain adaptation performance. Xie *et al.* [7] show that adversarial training can improve both standard accuracy and robustness against distribution shifts, with the help of auxiliary batch normalization. Adversarial Noise Training (ANT) [8] uses random Gaussian noise with learned adversarial hyperparameters (mean and variance) as the augmentation. Gowal *et al.* [54] propose to generate new augmented images by adversarially composing the representations of different original images. Wang *et al.* [55] improve the robustness of pose estimation models by adversarially mixing different corrupted images as data augmentation. Based on DeepAugment [3], Calian *et al.* [56] propose AdversarialAugment, which optimizes the parameters of image-to-image models to generate adversarially augmented images, achieving promising results against natural corruptions. Robey *et al.* [57] proposed model-based robust learning which generates augmented images with adversarial natural variations to improve model robustness against naturally-occurring conditions (e.g., snow, decolorization, shadow, and many others). The authors further generalized such idea to the domain generalization problem and achieved impressive results [58].

## 2.3 Normalization

Batch Normalization (BN) [59] has been a cornerstone for training deep networks. Inspired by BN, more task-specific modifications are proposed by exploiting different normalization axes, such as Instance Normalization (IN) [60], Layer Normalization (LN) [61] and Group Normalization (GN) [62]. A few latest works investigate to use multiple normalization layers instead of one BN. Zajkac *et al.* [63] use two separate BNs to handle the domain shift between labeled and unlabeled data in semi-supervised learning. Xie *et al.* [64, 7] observe the difference between standard and adversarial feature statistics during AT, and craft *dual batch normalization* (DuBN) to disentangle the standard and adversarial feature statistic to improve both the standard accuracy and robustness. While most deep models just use one normalization type throughout the network, IBN-Net [29] unifies IN and BN in one model. It finds that when applied together, IN tends to learn features invariant to appearance (colors, styles, *etc.*) changes, while BN is essential for preserving high-level semantic contents. IBN-Net shows that combining IN and BN in an appropriate manner can improve model generalization. A similar combination was also explored in the style transfer field [65], learning to selectively normalize only disturbing styles while preserving useful styles.

# 3 Method

## 3.1 Preliminary: The AugMix Framework

At a high level, AugMix creates a new composite image from a clean sample by first applying several simple augmentation operations to it before combining the results through random linear combinations. This augmentation scheme is coupled with a Jensen-Shannon consistency loss that enforces feature similarity among clean and augmented images. Specifically, AugMix consists of multiple augmentation chains (3 by default), each applied in parallel to the same input image. Each chain composes one to three randomly selected augmentation operations. These augmentations are the same as AutoAugment [21], but excluding those that overlap with ImageNet-C corruptions. The augmented images from each augmentation chains are then combined together with the original, clean sample using a linear combinations with randomly samples coefficients. The final image thus incorporates several sources of randomness, coming respectively from the choice of operations, the severity level of these operations, the lengths of the augmentation chains and the mixing weights. Despite its simplicity, AugMix achieves state-of-the-art corruption robustness. In the remaining of this section, we build on the previous AugMix work and propose AugMax, an augmentation strategy that further improves robustness.

## 3.2 AugMax: Augmented Training with Unified Diversity and Hardness

In this section, we rigorously introduce our new data augmentation method, AugMax, illustrated in Figure 2. At a high level, the main difference between AugMax and AugMix is a new optimization procedure used to select the mixing weights $m$ and $\boldsymbol{w}$.

Given a data distribution $\mathbb{D}$ over images $\boldsymbol{x} \in \mathbb{R}^d$ and labels $\boldsymbol{y} \in \mathbb{R}^c$, our goal is to train a mapping (classifier) $f : \mathbb{R}^d \to \mathbb{R}^c$ from images to output softmax probabilities, parameterized by $\boldsymbol{\theta}$, that is robust to (unknown) distribution shifts. We minimize the empirical risk

$$\min_{\boldsymbol{\theta}} \mathbb{E}_{(\boldsymbol{x},\boldsymbol{y})\sim\mathcal{D}} \, \mathcal{L}(f(\boldsymbol{x};\boldsymbol{\theta}), \boldsymbol{y}), \tag{1}$$

where $\mathcal{L}(\cdot, \cdot)$ is the loss function (*e.g.*, cross-entropy). As illustrated in Figure 2, an AugMax image $\boldsymbol{x}^*$ is generated by learning a set of adversarial mixing parameters $m^*, \boldsymbol{w}^*$ which maximizes the loss:

$$\boldsymbol{x}^* = g(\boldsymbol{x}_{orig}; m^*, \boldsymbol{w}^*), \tag{2}$$

where $g(\cdot)$ denotes the AugMax augmentation function, $\boldsymbol{x}_{orig}$ is the original image, and

$$m^*, \boldsymbol{w}^* = \arg\max_{m,\boldsymbol{w}} \mathcal{L}(f(g(\boldsymbol{x}_{orig}; m, \boldsymbol{w}); \theta), \boldsymbol{y}), \quad \text{s.t. } m \in [0,1], \boldsymbol{w} \in [0,1]^b, \boldsymbol{w}^T \mathbf{1} = 1. \tag{3}$$

To simplify the optimization problem in Eq. (3), we use a re-parameterization trick by setting $\boldsymbol{w} = \sigma(\boldsymbol{p})$, where $\sigma(\cdot)$ is the softmax function, and convert the optimization into a differentiable one:

$$m^*, \boldsymbol{p}^* = \arg\max_{m \in [0,1], \boldsymbol{p} \in \mathbb{R}^b} \mathcal{L}(f(g(\boldsymbol{x}_{orig}; m, \sigma(\boldsymbol{p})); \theta), \boldsymbol{y}); \quad \boldsymbol{w}^* = \sigma(\boldsymbol{p}^*) \tag{4}$$

Training with AugMax augmentation can then be written altogether as minimax optimization:

$$
\begin{aligned}
&\min_{\boldsymbol{\theta}} \mathbb{E}_{(\boldsymbol{x},\boldsymbol{y})\sim\mathcal{D}} \, \frac{1}{2}[\mathcal{L}(f(\boldsymbol{x}^*); \boldsymbol{\theta}), \boldsymbol{y}) + \mathcal{L}(f(\boldsymbol{x}); \boldsymbol{\theta}), \boldsymbol{y})] + \lambda \mathcal{L}_c(\boldsymbol{x}, \boldsymbol{x}^*) \\
&\text{s.t. } \boldsymbol{x}^* = g(\boldsymbol{x}; m^*, \boldsymbol{w}^*); \; \boldsymbol{w}^* = \sigma(\boldsymbol{p}^*); \; m^*, \boldsymbol{p}^* = \arg\max_{m \in [0,1], \boldsymbol{p} \in \mathbb{R}^b} \mathcal{L}(f(g(\boldsymbol{x}; m, \sigma(\boldsymbol{p})); \boldsymbol{\theta}), \boldsymbol{y})
\end{aligned} \tag{5}
$$

where $\mathcal{L}_c$ is a consistency loss regularizing augmented images to have similar model outputs with the original images and $\lambda$ is the trade-off parameter. Our implementation of $\mathcal{L}_c$ is adapted from [5]:

$$\mathcal{L}_c(\boldsymbol{x}, \boldsymbol{x}^*) = \text{JS}(f(\boldsymbol{x};\theta), f(\tilde{\boldsymbol{x}};\theta), f(\boldsymbol{x}^*;\theta)) \tag{6}$$

with JS$(\cdot)$ being the Jensen-Shannon divergence and $\tilde{\boldsymbol{x}}$ being the augmented image generated by AugMix from $\boldsymbol{x}$. In order to reduce the training complexity overhead caused by adversarial training, we employ an accelerated adversarial attack method [28] to solve Eq. (4), which adds only a reasonable amount of extra complexity compared to AugMix baseline. The algorithm to solve Eq. (5) is summarized in Appendix A.

**Visualization of the effects of diversity and hardness**   To illustrate the motivation behind Aug-Max, we visualize the feature representations induced by different augmentation methods to showcase their diversity and hardness in Figure 1. Specifically, we visualize the impact of various data augmentation methods on the feature representations obtained with a ResNeXt29 that is normally trained on CIFAR-10. Note that the model is trained using only default standard augmentations (*i.e.*, random flipping and translation), and therefore has never seen augmentation from AugMix, AugMax, or PGD attack. Following [4], we visualize the penultimate layer feature distributions of training samples from three different classes. Three hundred training images are selected from each class for visualization. For AugMix and $\ell_\infty$ PGD attack, we use the same hyperparameters in the original papers [5, 6].

We observe in Figure 1 that the clean images form well-separated clusters in the feature space, leaving large regions between those clusters empty. This potentially causes uncertain and unreliable generalization if new test samples (from a shifted distribution) fall into those regions. AugMix enlarges each cluster to cover a broader region, effectively increasing sample diversity. However, after augmentation, very few samples are close to the decision boundaries, revealing an inability to generate hard samples. Samples from the PGD attack, on the other hand, collapse into a small region around the classification borders, losing feature diversity. We further investigate the method through ablation studies in Section 4.3.

### 3.3   DuBIN: Disentangled Normalization for Heterogeneous Features

AugMax leads to better coverage of the input space by unifying sample diversity and hardness, as demonstrated in Figure 1. While such more versatile distribution in data augmentation has the potential to increase robustness, we do observe that *naive* incorporation of AugMax leads to relatively marginal improvement over AugMix (see Section 4.3). This is due to the comprehensiveness of AugMax can also lead to a higher feature heterogeneity, which may require larger model capacity to encode. To address this problem, we design a novel normalization layer, termed *Dual Batch-and-Instance Normalization* (**DuBIN**), to disentangle the instance-level heterogeneity.

As illustrated in Figure 3, DuBIN consists of two parallel parts: (1) a dual batch normalization (DuBN) [7], which is the **by-default** normalization layer to use for adversarial training [7, 64], to disentangle the group-level statistics of the clean and corner-case augmented samples[2]; and (2) an additional instance normalization, to account for instance-level feature statistics due to augmentation diversity. At each layer, the input feature is split into halves along the channel dimension: one half is then fed into IN, and the other fed into DuBN. Finally, the outputs of IN and DuBN are concatenated back along the channel dimension for the next layer.

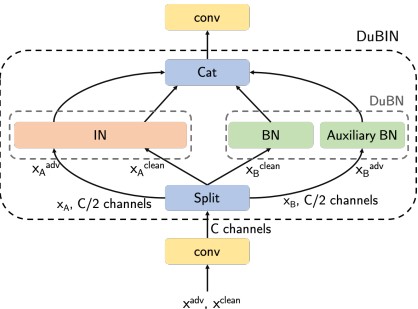

Figure 3: Illustration of DuBIN. "Split" and "cat" represent splitting and concatenating on features along the channel dimension. The input feature $x$ with $C$ channels is split into halves (i.e., $x_A$ and $x_B$, each with $C/2$ channels). Each half goes into different normalization layers (either IN or DuBN) and the outputs are concatenated back along the channel dimension.

To provide additional insights of DuBIN, we compare the BN statistics of AugMax-DuBN and AugMax-DuBIN in Table 1. $\overline{\sigma}_c^2$ and $\overline{\sigma}_a^2$ represent the average batch normalization variance over all channels of $BN_c$ and $BN_a$[2], respectively. We observe that both $\overline{\sigma}_c^2$ and $\overline{\sigma}_a^2$ are smaller in the DuBIN network than those in the DuBN counterpart. This shows that the IN branch in DuBIN can reduce the feature diversity that BN otherwise needs to model, by encoding instance-level diversities. With lower feature variation, BNs can converge better with improved performance [66].

Table 1: BN statistics of different layers in WRN40-2 with DuBN or DuBIN trained on CI-FAR100 using AugMax training.

| Layer | | Block 1 layer 2 | Block 1 layer 3 | Block 2 layer 2 |
|---|---|---|---|---|
| AugMax-DuBN | $\overline{\sigma}_c^2$ | 0.0369 | 0.0450 | 0.0301 |
| | $\overline{\sigma}_a^2$ | 0.0469 | 0.0585 | 0.0382 |
| AugMax-DuBIN | $\overline{\sigma}_c^2$ | 0.0306 | 0.0403 | 0.0264 |
| | $\overline{\sigma}_a^2$ | 0.0348 | 0.0466 | 0.0292 |

---

[2]DuBN consists of two separate BNs: one $BN_c$ for clean images and the other $BN_a$ for adversarially augmented (e.g., AugMax) images.

# 4 Experiments

Here, we introduce in detail the experimental setting, and report quantitative results for robustness against natural corruptions and distribution shifts. We show that our method induces better robustness than existing ones. We study the properties of our approach in thorough ablation studies.

## 4.1 Experimental Setup

**Datasets and Models** We evaluate our proposed method on CIFAR10, CIFAR100 [31], ImageNet [32] and Tiny ImageNet (TIN)[3]. For neural architectures, we use a ResNet18 [67], WRN40-2 [68] and ResNeXt29 [69] on the CIFAR datasets, and ResNet18 on ImageNet and Tiny ImageNet[4]. To evaluate the model's robustness against common natural corruptions, we use CIFAR10-C, CIFAR100-C, ImageNet-C and Tiny ImageNet-C (TIN-C) [1], which are generated by adding natural corruptions to the original test set images.

**Evaluation Metrics** Following established conventions [1, 5], we define robustness accuracy (RA) as the average classification accuracy over all 15 corruptions, to measure the models' robustness on CIFAR10-C and CIFAR100-C. On ImageNet-C and Tiny ImageNet-C, we use both RA and mean corruption error (mCE) to evaluate robustness. As defined by [1], mCE is the weighted average of target model corruption

Table 2: Evaluation results on CIFAR10 and CIFAR10-C. The best metric is shown in bold.

| Model | Metric | Normal | AugMix | AugMax-DuBIN |
|-------|--------|--------|--------|--------------|
| ResNet18 | SA (%) | 95.56 | **95.79** | 95.76 (-0.03) |
| | RA (%) | 74.75 | 89.49 | **90.36** (+0.87) |
| WRN40-2 | SA (%) | 94.78 | 95.67 | **95.68** (+0.01) |
| | RA (%) | 73.71 | 89.01 | **90.67** (+1.66) |
| ResNeXt29 | SA (%) | 95.60 | 96.25 | **96.39** (+0.14) |
| | RA (%) | 71.70 | 89.08 | **92.11** (+3.03) |

errors normalized by the corruption errors of a baseline model across different types of corruptions. We use the AlexNet provide in [1] as the baseline model for experiments on ImageNet. Since [1] does not provide an off-the-shelf baseline model for Tiny ImageNet, we use the conventionally trained ResNet18 on Tiny ImageNet as the baseline model for experiments on Tiny ImageNet. We use standard accuracy (SA) to denote the classification accuracy on the original clean testing images.

**Baseline Methods** On CIFAR10-C and CIFAR100-C, we compare our method with the state-of-the-art method, AugMix [5]. In addition, AugMix has previously been combined with other orthogonal methods, such as DeepAugment [3], to further boost performance on ImageNet-C. Specifically, [3] combine DeepAugment as an orthogonal component to AugMix, and achieve state-of-the-art

Table 3: Evaluation results on CIFAR100 and CIFAR100-C. The best metric is shown in bold.

| Model | Metric | Normal | AugMix | AugMax-DuBIN |
|-------|--------|--------|--------|--------------|
| ResNet18 | SA (%) | 77.99 | 78.23 | **78.69** (+0.46) |
| | RA (%) | 48.46 | 62.67 | **65.75** (+3.08) |
| WRN40-2 | SA (%) | 76.19 | **77.03** | 76.80 (-0.23) |
| | RA (%) | 46.80 | 64.56 | **66.35** (+1.79) |
| ResNeXt29 | SA (%) | 79.95 | 78.58 | **80.70** (+2.12) |
| | RA (%) | 47.76 | 65.37 | **68.86** (+3.49) |

performance on ImageNet-C with DeepAugment + AugMix [3]. To allow comparison with this scenario, we also run the following comparison experiment on ImageNet-C and Tiny ImageNet-C: (1) AugMix v.s. AugMax and (2) DeepAugment + AugMix v.s. DeepAugment + AugMax. We show that the advantages of AugMax still exist when combined with other orthogonal methods.

Finally, we also compare AugMax with ANT [8], which utilizes Gaussian noises with learned distribution parameters for data augmentation. Since models tend to generalize better on test images similar to those seen during training [70], ANT mainly improves robustness against high frequency corruptions (*e.g.*, additive noises), while achieving comparable robustness on other unseen corruptions (*e.g.*, blurring) with AugMix. Following [8], we perform the comparison on the subset of Tiny ImageNet-C with 12 remaining corruptions after removing all additive noises.

Table 4: Evaluation results on ImageNet and ImageNet-C. The best metric is shown in bold.

| Method | SA (%, ↑) | RA (%, ↑) | mCE (%, ↓) |
|--------|-----------|-----------|------------|
| Normal | 69.83 | 30.91 | 87.47 |
| AugMix | **68.06** | 34.58 | 83.08 |
| AugMax-DuBIN | 67.62 (-0.44) | **35.01** (+0.43) | **82.56** (-0.52) |
| DeepAugment + AugMix | **65.32** | 45.84 | 69.29 |
| DeepAugment + AugMax-DuBIN | 64.43 (-0.89) | **46.55** (+0.71) | **68.47** (-0.82) |

---

[3]https://www.kaggle.com/c/tiny-imagenet
[4]Our WRN40-2 and ResNeXt29 models are identical to those used in AugMix [5].

**Implementation Details**   We follow [5] for hyperparameter settings. Specifically, for all experiments on CIAFR10 and CIFAR100, we use SGD optimizer with initial learning rate 0.1 and cosine annealing learning rate scheduler, and train all models for 200 epochs. For all experiments on ImageNet, we use SGD optimizer with initial learning rate 0.1 and batch size 256 to train the model for 90 epochs. We reduce the learning rate by $1/10$ at the 30-th and 60-th epoch. For all experiments on Tiny ImageNet, we train for 200 epochs and decay the learning rate at the 100-th and 150-th epoch. Other hyperparameter settings are identical as those in ImageNet experiments.

We set batch size to 256 for all experiments. In Eq. (4), $m$ and $\boldsymbol{p}$ are uniformly randomly initialized between 0 and 1. We use PGD to update $m$: first do gradient ascend on $m$ and then project it back into the $[0, 1]$ interval. $\boldsymbol{p}$ is updated using gradient ascend. AugMax follows the same rules as AugMix [5] to randomly select augmentation operation type, severity level and chain length. Following [6], we iteratively solve the inner maximization and outer minimization problems in Eq. (5): the inner maximization is updated for $n$ steps, for every 1 step update on the outer minimization. Both $m$ and $\boldsymbol{p}$ are updated with step size $\alpha = 0.1$. We set $n = 5$ on ImageNet for efficiency and $n = 10$ on other datasets. We set $\lambda$ in Eq. (5) to be 12 on ImageNet and 10 on all other datasets, except for ResNet18 on CIFAR100 where we find $\lambda = 1$ leads to better performance. All experiments are conducted on a server with four NVIDIA RTX A6000 GPUs.

Table 5: Evaluation results on TIN and TIN-C. The best metric is shown in bold.

| Method | SA (%, ↑) | RA (%, ↑) | mCE (%, ↓) |
|---|---|---|---|
| Normal | 61.64 | 23.91 | 100.00 |
| AugMix | 61.79 | 36.85 | 83.04 |
| AugMax-DuBIN | **62.21** (+0.42) | **38.67** (+1.82) | **80.72** (-2.32) |
| DeepAugment + AugMix | 59.59 | 40.67 | 78.28 |
| DeepAugment + AugMax-DuBIN | **59.72** (+0.13) | **40.99** (+0.32) | **77.83** (-0.45) |

## 4.2   Robustness against Natural Corruptions

Results on CIFAR10-C and CIFAR100-C are shown in Table 2 and 3 respectively, where "normal" means training with default standard augmentations (*e.g.*, random flipping and translation). From the results, we see that AugMax-DuBIN achieves new state-of-the-art performance on both datasets with different model structures. For example, compared with AugMix, our method improves accuracy by as high as 3.03% and 3.49% on CIFAR10-C and CIFAR100-C, respectively. Moreover, we observe that our method benefits larger models more. Specifically, in both CIFAR10 and CIFAR100 experiments, the largest robustness gain is achieved on ResNeXt29, which has the largest capacity among the three models.

Results on ImageNet are shown in Table 4. AugMax-DuBIN outperforms AugMix by 0.52% in terms of mCE on ImageNet-C. Further combining AugMax-DuBIN with DeepAugment outperforms AugMix + DeepAugment by 0.82% in terms of mCE, achieving a new state-of-the-art performance on ImageNet-C. This

Table 6: Evaluation results on TIN and TIN-C (w/o noise). The best metric is shown in bold.

| Method | SA (%, ↑) | RA (%, ↑) | mCE (%, ↓) |
|---|---|---|---|
| Normal | 61.64 | 24.38 | 100.00 |
| AugMix | 61.79 | 37.63 | 82.54 |
| ANT | 61.26 | 35.30 | 85.70 |
| AugMax-DuBIN | **62.21** | **38.66** | **80.29** |

shows that AugMax can be used as a more advanced basic building block for model robustness and inspires future researches to build other defense methods on top of it.

Results on Tiny ImageNet-C are shown in Table 5 and Table 6. From Table 5, we see that our method improves the mCE by 2.32% compared with AugMix and 0.45% when combined with DeepAugment. From Table 6, we can see our method outperforms ANT by a considerable margin.

**Training Time**   Since we use an accelerated adversarial training method [28] to solve Eq. (4), the worst-case mixing strategy can be searched efficiently. As a result, AugMax adds only a modest training overhead over AugMix, and is significantly more efficient than traditional adversarial training (AT). Detailed training time is shown in Table 7.

Table 7: Training time on ImageNet with ResNet18, reported on a single NVIDIA A6000.

| Method | Time (sec/epoch) |
|---|---|
| Normal | 2669 |
| AugMix | 3622 |
| AT [6] | 37162 |
| AugMax | 5264 |

## 4.3   Ablation Study

**Analysis of Different Normalization Layers**   In this paragraph, we compare AugMax/AugMix when combined with different normalization schemes. When using DuBN or DuBIN, we route

original images to the $BN_c$, and AugMax/AugMix images to $BN_a$. Since it is the by-default setting to use disentangled normalization layers (e.g., DuBN) for original and augmentation images in adversarial training [7, 64], we follow this traditional routing and combine AugMax with DuBN or DuBIN instead of BN or IBN. It is fair to compare AugMax-DuBN/DuBIN with either AugMix-BN/IBN or AugMix-DuBN/DuBIN.

Results are shown in Table 8. AugMax-DuBN outperforms both AugMix-BN and AugMix-DuBN, and AugMax-DuBIN outperforms both AugMix-IBN and AugMix-DuBIN. Noticeably, when combined with AugMix, DuBIN also helps improve robustness. This shows the potential of applying DuBIN to other diversity augmentation methods for general performance improvement.

Table 8: Ablation results on DuBIN. RA (%) on CIFAR10/100-C with WRN40-2 backbone are reported. Mean and standard derivation over three random seeds are shown for each experiment.

| Method | CIFAR10-C | CIFAR100-C |
|---|---|---|
| AugMix-BN | 89.01 ($\pm$ 0.03) | 64.56 ($\pm$ 0.04) |
| AugMix-IBN | 89.17 ($\pm$ 0.24) | 63.94 ($\pm$ 0.28) |
| AugMix-DuBN | 89.11 ($\pm$ 0.21) | 64.07 ($\pm$ 0.38) |
| AugMix-DuBIN | 89.74 ($\pm$ 0.35) | 65.02 ($\pm$ 0.58) |
| AugMax-DuBN | 89.60 ($\pm$ 0.62) | 65.06 ($\pm$ 0.28) |
| AugMax-DuBIN | **90.67** ($\pm$ 0.16) | **66.35** ($\pm$ 0.21) |

We also conduct stability analyses on these experiments, by showing the statistical significance of the improvements achieved by AugMax-DuBIN over the algorithm randomness. Specifically, we run all experiments in Table 8 using three different random seeds, and report the mean (denoted as $\mu$) and standard derivation (denoted as $\sigma$) of accuracy in the form of $\mu (\pm \sigma)$ in Table 8. As we can observe, the improvements achieved by AugMax-DuBIN are statistically significant and consistent.

**AugMax Hyperparameters** In this paragraph, we check the sensitivity of AugMax with respect to its hyperparameters: maximization steps $n$, step size $\alpha$, and consistency loss tradeoff parameter $\lambda$. The results are shown in Table 9. We first fix $\lambda = 10$ and try different values for $n$ and $k$. We fix $k = 1$ when adjust-

Table 9: Ablation results on AugMax-DuBIN update step number $n$ and, early-stopping step $k$ and consistency loss tradeoff parameter $\lambda$. Results are reported on CIFAR10/CIFAR10-C with WRN40-2.

| | $n$ | | $k$ | | | $\lambda$ | | |
|---|---|---|---|---|---|---|---|---|
| | 5 | 10 | 1 | 2 | 3 | 1 | 10 | 15 |
| SA (%) | 95.78 | 95.68 | 95.68 | 95.60 | 95.51 | 95.90 | 95.68 | 95.88 |
| RA (%) | 90.49 | 90.67 | 90.67 | 90.38 | 90.21 | 88.89 | 90.67 | 90.30 |

ing $n$ and fix $n = 10$ when adjusting $k$. As we can see, AugMax is stable with respect to different values of $n$ and $k$. We use $n = 10$ and $k = 1$ as the default values since they empirically yield good results at low training overhead. We then fix $n = 10, k = 1$ and adjust $\lambda$. We find robustness peaks at around $\lambda = 10$, which we set as the default value.

**Comparison with Different Diversity and Hardness Strategies** In this paragraph, we study different strategies to combine diversity and hardness. The results confirm that diversity and hardness are two complementary dimensions and their proper unification boosts model robustness.

Specifically, we compare AugMax with baseline methods from each strategy group (*i.e.*, diversity or hardness), and other possible strategies to combine both. Besides AugMix and Adversarial Training (e.g., PDGAT [6], FAT [28]), which are the two representative examples for diversity and hardness as shown in Figure 1, we also design the following baseline methods. (1) AugMix+PGDAT: using both AugMix and adversarial images for augmentation. This is a naive baseline to combine diversity with hardness. (2)

Table 10: Results of different augmentation strategies on CIFAR100(-C) with ResNet18.

| Method | Strategy | SA (%) | RA (%) |
|---|---|---|---|
| Normal | - | 77.99 | 48.46 |
| AugMix [5] | (diversity) | 78.23 | 62.67 |
| PGDAT [6] | | 60.94 | 47.71 |
| FAT [28] | (hardness) | 61.51 | 48.70 |
| AdvMax | | 56.61 | 38.65 |
| AugMix+PGDAT | | 61.68 | 51.39 |
| AdvMix | (diversity & adversity) | 72.36 | 56.89 |
| AugMax-DuBIN | | **78.52** | **64.02** |

AdvMix: applying adversarial attacks on the augmentation hyperparameters such as the rotation angles and translation pixel numbers [71, 72], while randomly selecting the mixing parameters; (3) AdvMax: applying adversarial attacks on both the augmentation hyperparameters and the mixing weights.[5] Please see Appendix B for their implementation details.

---

[5]Note that AdvMax implies full hardness and little diversity, while AdvMix switches the order of randomness and adversarial augmentation in the AugMax pipeline.

The results are shown in Table 10. AugMax-DuBIN achieves the best performance, outperforming all methods from either diversity or hardness group. This shows diversity and hardness to be indeed complementary. On the other hand, the other two naive baselines jointly considering diversity and hardness achieve poorer performance, showing it nontrivial to design a method achieving good balance between diversity and hardness.

### 4.4 Robustness against Other Distribution Shifts

Although AugMax mainly aims at improving model robustness against common corruptions, we find that it can also gain robustness against other types of distribution shifts. Specifically, we evaluate on the distribution shifts caused by the differences in data collection process using CIFAR10.1 [30], and against Spatial Transform adversarial Attacks (STA) [71] on CIFAR10-STA. We generate CIFAR10-STA by applying the worst-of-$k$ attack with $k = 10$ on CIFAR10 test set following [71].

Table 11: Robustness against other distribution shifts. Accuracy (%) on CIFAR10.1 and CIFAR10-STA are evaluated on ResNeXt29 trained on CIFAR10.

| Method | CIFAR10.1 | CIFAR10-STA |
|---|---|---|
| Normal | 88.90 | 30.30 |
| AugMix | 88.90 | 54.30 |
| AugMax-DuBIN | **90.64** | **63.20** |

Results are reported in Table 11 (using the same ResNeXt29 models trained on CIFAR10 as in Section 4.2). AugMax-DuBIN steadily outperforms AugMix against both distributional shifts.

## 5 Conclusion

In this paper, we propose AugMax, a novel data augmentation strategy, that significantly improves robustness by strategically unifying diversity and hardness. To enable efficient training while facing the resulting heterogeneous features, we design a novel normalization scheme termed DuBIN. Using the combination of AugMax and DuBIN, we consistently demonstrate state-of-the-art robustness on several natural corruption benchmarks.

AugMax unifies diversity and hardness in a heuristic manner. While we showed in ablation studies that AugMax outperforms other heuristic combinations, there may be better trade-offs available. We leave such study for future work and focus in this paper on showing the benefit of unifying diversity and hardness, which were separately considered in previous research.

## Acknowledgement

Z.W. is supported by the U.S. Army Research Laboratory Cooperative Research Agreement W911NF-17-2-0196 (IOBT REIGN), and an NVIDIA Applied Research Accelerator Program.

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
