# A AugMax Algorithm

The algorithm to generate AugMax images from clean images (*i.e.*, to solve Eq. (4)) is summarized in Algorithm 1, where we employ an accelerated adversarial attack method [28] to reduce complexity. The basic idea behind the acceleration is to early-stop gradient ascend when misclassification has already occurred in $k$ iterations. For all our experiments, we use $k = 1$, $n = 5$ and $\alpha = 0.1$.

---

**Algorithm 1:** Generate AugMax Images

---

**Input:** Original image $\boldsymbol{x}$, one-hot label vector $\boldsymbol{y}$, early-stopping step $k$, maximum step $n$, step size $\alpha$, AugMax augmentation function $g(\cdot)$, classifier $f(\cdot)$ with parameter $\boldsymbol{\theta}$, loss function $\mathcal{L}(\cdot, \cdot)$.

**Output:** AugMax image $\boldsymbol{x}^*$

1   Randomly initialize $m^* \in [0, 1]$ and $\boldsymbol{p}^* \in \mathbb{R}^b$.

2   $\boldsymbol{w}^* \leftarrow \sigma(\boldsymbol{p}^*)$    // $\sigma(\cdot)$ is softmax function.

3   $\boldsymbol{x}^* \leftarrow g(\boldsymbol{x}; m^*, \boldsymbol{w}^*)$

4   $c \leftarrow 0$

5   **for** $i \leftarrow 1$ **to** $n$ **do**

6      $m^* \leftarrow m^* + \alpha \text{sign}(\nabla_{m^*} \mathcal{L}(f(\boldsymbol{x}^*; \boldsymbol{\theta}), \boldsymbol{y}))$    // Gradient ascend on $m^*$.

7      $m^* \leftarrow \text{clip}(m^*, 0, 1)$

8      $\boldsymbol{p}^* \leftarrow \boldsymbol{p}^* + \alpha \text{sign}(\nabla_{\boldsymbol{p}^*} \mathcal{L}(f(\boldsymbol{x}^*; \boldsymbol{\theta}), \boldsymbol{y}))$    // Gradient ascend on $\boldsymbol{p}^*$.

9      $\boldsymbol{w}^* \leftarrow \sigma(\boldsymbol{p}^*)$

10     $\boldsymbol{x}^* \leftarrow g(\boldsymbol{x}; m^*, \boldsymbol{w}^*)$

11     **if** $\arg\max_j f(\boldsymbol{x}^*) \neq \arg\max_j \boldsymbol{y}$ **then**

12       $c \leftarrow c + 1$

13     **end**

14     **if** $c = k$ **then**

15       **break**    // Early stopping.

16     **end**

17 **end**

---

**Algorithm 2:** Robust Learning with AugMax

---

**Input:** Dataset $\mathcal{D}$, iteration number $T$, batch size $B$, classifier $f(\cdot)$, loss functions $\mathcal{L}(\cdot, \cdot)$ and $\mathcal{L}_c(\cdot, \cdot)$, learning rate $\rho$, loss trade-off parameter $\lambda$.

**Output:** Learned parameter $\boldsymbol{\theta}$ for $f(\cdot)$.

1   Randomly initialize $\boldsymbol{\theta}$.

2   **for** $t \leftarrow 1$ **to** $T$ **do**

3     Sample images and corresponding labels $\{(\boldsymbol{x}_i, \boldsymbol{y}_i)\}_{i=1}^B$ from $\mathcal{D}$.

4     **for** $i \leftarrow 1$ **to** $B$ **do**

5       Generate AugMax image $\boldsymbol{x}_i^*$ from $\boldsymbol{x}_i$ following Algorithm 1.

6     **end**

7     $\boldsymbol{\theta} \leftarrow \boldsymbol{\theta} - \rho \nabla_{\boldsymbol{\theta}} \frac{1}{B} \sum_{i=1}^B \{ \frac{1}{2}[\mathcal{L}(f(\boldsymbol{x}_i^*; \boldsymbol{\theta}), \boldsymbol{y}_i) + \mathcal{L}(f(\boldsymbol{x}_i); \boldsymbol{\theta}), \boldsymbol{y}_i)] + \lambda \mathcal{L}_c(\boldsymbol{x}_i, \boldsymbol{x}_i^*) \}$

8 **end**

---

The algorithm to train a robust classifier with AugMax images (*i.e.*, to solve Eq. (5)) is summarized in Algorithm 2.

# B Details on AdvMix and AdvMax

For AdvMix and AdvMax, we use the worst-of-$k$ method [71] to do adversarial attack on the augmentation hyperparameters such as rotation angles and translation pixel numbers, where $k$ is set to 5. Specifically, for AdvMix, we first randomly select augmentation operations types and mixing parameters as done in AugMix. We then randomly sample $k$ sets of augmentation hyperparameters from the allowed intervals predefined in [5]. We then use the one leading to largest classification loss to generate AdvMix images. For AdvMax, we first follow the same routine as AdvMix to generate worst-case augmentation hyperparameters, and then use the same way to learn worst-case mixing parameters as AugMax. In order to further explore the hard-cases, we also include a stronger spatial transform attack, StAdv [72], in AdvMax.