# OpenReview forum: "AugMax: Adversarial Composition of Random Augmentations for Robust Training"
_NeurIPS.cc/2021/Conference — NeurIPS 2021 Poster_

### Official Review · Reviewer_Gx5n · 2021-07-07

**Rating:** 7
**Confidence:** 3

**Summary:**

This paper proposed AugMax, which increases both the diversity and the hardness. To make the training easier, the paper design a normalization strategy DuBIN.  Experimental results show that the combination of AugMax and DuBIN exhibits good performance.

**Limitations And Societal Impact:**

Please refer to Main Review.

**Main Review:**

The paper can inspire future AT methods to deal with OOD samples. The idea is intuitive and easy to follow.

My concerns:
1. I am confused by the DuBIN layer and not sure how it works, can you show me the formulations of the 2 normalizations?
2.
2. In Table 10, you reported the performance of AT on CIFAR100. Can your method be extended to the AT settings, i.e., can your method defend the adversarial examples.
3. You state that "AugMax can also lead to a higher feature heterogeneity, which may require larger model capacity to encode." Have you tried Different Normalization Layers on different models (e.g., resnet, wrn, etc.) to show how model capacity affects the performance.



**Time Spent Reviewing:**

3

---

> ### Author Response · Authors · 2021-08-10
> **Response to reviewer Gx5n**
>
> Q: What are the formulations of DuBIN?
>
> A: Thank you for your comment, we will add the formulations to the updated manuscript. As described in lines 207-214, DuBIN consists of two parallel parts: a DuBN and an IN. At each layer, the input feature is split into halves along the channel dimension: one half is then fed into IN, and the other fed into DuBN. Finally, the outputs of IN and DuBN are concatenated back along the channel dimension for the next layer.
>
> DuBN, as the default normalization layer in adversarial training [7, 45], has two separate batch normalizations, namely BNc and BNa.
>
> During training, we route normal samples to BNc and adversarially augmented samples (e.g., AugMax samples) to BNa. So the formulations are:
>
> BNc: $\gamma_c * (x_{normal} - \mu_c) / \sigma_c + \beta_c$
>
> BNa: $\gamma_a * (x_{augmax} - \mu_a) / \sigma_a + \beta_a$
>
> where $\mu$ and $\sigma$ are the mean and standard deviation, $\gamma$ and $\beta$ are the affine transformation parameters, subscripts $c$ and $a$ indicate which BN they belong to, and $x_{normal}$ and $x_{augmax}$ are the feature tensors for normal and augmax training samples respectively.
>
> During inference, we route all test samples to BNc.
>
> Please check [7,45] for more details on DuBN.
>
> Q: Can AugMax defense adversarial attacks?
>
> A: AugMax empirically achieves some level of robustness against adversarial attacks, although it is designed to defend against natural corruptions instead of adversarial attacks.
> For example, on CIFAR10 with WRN40-2 backbone, the accuracy against spatial transform attacks (solved by the worst-of-k method with k=10, rotation limit 30 degrees and translation limit 3 pixels) [52] are shown below.
>
> Normal: 27.07%
>
> AugMix: 48.22%
>
> AugMax-DuBIN: 56.76%
>
> As we can see, AugMax-DuBIN has better adversarial robustness than AugMix.
>
> Q: Try different models to show how model capacity affects the performance.
>
> A: Results on the WRN40 backbone with two different widening factors (2, 5) and normalization layers (DuBN and DuBIN) on CIFAR10 are shown below:
>
> WRN40-2, AugMax-DuBN: SA=95.65%, RA=89.60%
>
> WRN40-5, AugMax-DuBN: SA=96.26%, RA=90.73%
>
> WRN40-2, AugMax-DuBIN: SA=95.68%, RA=90.67%
>
> WRN40-5, AugMax-DuBIN: SA=96.41%, RA=91.83%
>
> As we can see, models with larger widening factors (and thus larger model capacity) have better performance, and DuBIN outperforms DuBN under the same widening factors.

---

### Official Review · Reviewer_fdjn · 2021-07-10

**Rating:** 6
**Confidence:** 3

**Summary:**

This paper proposes a novel data augmentation method AugMax which adversarially learns the parameters for integrating randomly selected augmentation operators to achieve a unification of diversity and hardness. To solve the issue of feature heterogeneity incurred by AugMax, this paper proposes DuBIN which simultaneously uses DuBN and IN instead of only DuBN. AugMax-DuBIN can improve OOD robustness while almost maintaining generalization.

**Ethical Concerns:**

No ethical issues.

**Limitations And Societal Impact:**

Limitations: Please carefully refer to "Cons" in Main Review.
No potential negative societal impact.

**Main Review:**

Pros:
1.	This paper is well organized. The motivation of AugMax-DuBIN is clear. AugMax utilizes the strengths of both AugMix and PGD attack to gain a unification of diversity and hardness. DuBIN combines IN with DuBN to reduce the feature diversity which is validated by smaller batch normalization variance brought by BuBIN.
2.	Observing from the experimental results, AugMax-DuBIN can significantly improve robust accuracy and decrease mean corruption error.

Cons:
1.	Please illustrate the differences between robustness accuracy (RA) and mCE Since RA is also evaluated as the average accuracy on corruptions. In addition, please provide more details about the corruption methods.
2.	It is better to provide the ablation study on the effect of early-stopping step $k$ in Algorithm 1. I suppose it is a critical hyper-parameter for the performance of AugMax-DuBIN. In addition, please illustrate the differences between AugMax with $k=10$ and PGD with 10 iteration steps. Also, it is better to conduct a fair comparison between FAT[1] which utilizes early-stepped PGD to improve robustness.
3.	This paper lacks some theoretical analyses to further validate the efficacy of AugMax.
4.	It is a bit wired to claim achieve a unification of diversity and hardness. Figure 1 illustrates AugMax gains a trade-off between diversity and hardness. It seems difficult to obtain enough diversity and enough hardness simultaneously. Please provide some discussions about this concern.

[1] Attacks which do not kill training make adversarial learning stronger. ICML 2020


**Time Spent Reviewing:**

3

---

> ### Author Response · Authors · 2021-08-10
> **Response to reviewer fdjn**
>
> Q: What is the difference between RA and mCE?
>
> A: Thank you for your comment, we will add this clarification to the updated manuscript. mCE is the weighted average of classification errors against different corruptions. It normalizes classification errors against different corruptions by dividing by AlexNet’s errors. This is to make error rates against different corruptions more comparable since different corruptions pose different levels of difficulty. (Please refer to [1] for more details.) In contrast, RA is the unweighted average of accuracy on different corruptions. Following previous works [1,3], we use RA on small-scale datasets such as CIFAR10-C and CIFAR100-C, and use both RA and mCE on large-scale datasets such as ImageNet-C.
>
>
> Q: Please provide more details about the corruption methods.
>
> A: We consider 15 common natural corruptions provided in [1]. More specifically, the 15 corruptions are drawn from four main categories: noises (Gaussian noise, shot noise, impulse noise), blurrings (defocus blur, frosted glass blur, motion blur, zoom blur), bad weathers (snow,  frost, fog) and digital distortions (brightness, contrast, elastic, pixelate, JPEG). Please refer to [1] for more details and visualizations. We will add these details in the revision.
>
> Q: Ablation study on different k values in AugMax-DuBIN.
>
> A: We conducted this ablation study on CIFAR10 with a WRN40-2 backbone based on the reviewer’s suggestion. The results are shown below.
>
> k=1: SA=95.68%, RA=90.67%
>
> k=2: SA=95.60%, RA=90.38%
>
> k=3: SA=95.51%, RA=90.21%
>
> As we can see, the performance of AugMax-DuBIN is not very sensitive to k, and our default k =1 yields good results at low training overhead. We will add these results in the revision.
>
>
> Q: What’s the difference between AugMax with k=10 and PGD with 10 iterations?
>
> A: AugMax has two iteration hyper-parameters: the early-stopping iteration k and the total iteration number n. If we set k=n=10 in AugMax, then no early-stopping is conducted. In this case, both AugMax and PGD do gradient ascend for 10 iterations. The difference is that AugMax optimizes the mixing parameters while PGD attack optimizes additive noise values.
>
> Q: Add experiments to compare with FAT.
>
> A: The results of FAT on CIFAR10 with WRN40-2 are shown below. We will add them in revision per your suggestion.
>
> FAT: SA=88.59%, RA=81.40%
>
> AugMax-DuBIN: SA=95.68%, RA=90.67%
>
> As we can see, AugMax-DuBIN outperforms FAT.
>
> Q: Theoretical analysis
>
> A: Similar to several previous works such as [3,5], our current work aims at providing an effective empirical solution to improve model robustness against natural corruptions, and the proposed method achieves state-of-the-art performance. We have also analyzed the insight of the proposed method. We agree that a more theoretical analysis will be interesting, and will leave it to our future work.
>
> Q: Add more discussions on diversity and hardness unification.
>
> A: We agree with you that AugMax essentially seeks a trade-off between diversity and hardness. Our point is to claim that AugMax achieves better trade-off between diversity and hardness compared to previous data augmentation methods, which focused either only on diversity or only on hardness. We will clarify this point in the revision.

---

### Official Review · Reviewer_iAMx · 2021-07-16

**Rating:** 6
**Confidence:** 5

**Summary:**

This paper presents an interesting new data augmentation called AugMax. It is motivated by diversity-hardness tradeoff of data augmentations. The authors suggest they are two complementary dimensions to achieve out-of-distribution robustness, and a disentangled normalization module termed DuBIN (Dual-Batch-and-Instance Normalization) could alleviate the resultant training difficulty.

**Ethical Concerns:**

No concern

**Limitations And Societal Impact:**

No concern was found. This paper should have positive social impact.

**Main Review:**

Pros:
-	Strong and clear motivation. The authors took a unified view towards two existing categories of data augmentation: random and adversarial. Either of the two was shown to improve domain shift robustness, but their complementariness was not previously revealed. The authors are the first to show a unification between the two is necessary to achieve robustness
-	Solid technical approach. The authors first proposed AugMax to reconcile diversity with hardness by learning an adversarially weighted combination of multiple random augmentations. Since this stronger augmentation leads to more diverse input distribution that is harder to fit, they further designed a new normalization strategy called DuBIN to disentangle the instance-wise feature heterogeneity of AugMax samples.
-	State of the art experimental results. Their simple approach yields significantly improved OoD robustness, outperforming strong previous methods by 2.01%, 2.52%, 1.84% and 15 2.13% on CIFAR10-C, CIFAR100-C, Tiny ImageNet-C and ImageNet-C.
-	Convincing ablation experiments. The authors compared different Diversity and Hardness Strategies The results confirm that diversity and hardness are two complementary dimensions, and their proper unification strategy is superior to alternative.
-	Writing is very clear and easy to follow.

Cons:
-	I am not fully convinced why AdvMix is not as good as AugMax in Table 10. Both seem to be sensible options to me to combine diversity and hardness. The author also did not provide an insight why.
-	In Table 8, DuBIN seems to boost both AugMix and AugMax, at similar margins over DuBN. That sets doubts on whether the authors’ motivation of adopting DuBIN is really the case, e.g., because AugMax introduces a particularly heterogenous input distribution.
-	AugMax looks computationally heavy, with default n = 10. There was also no mention of training overhead. Please address this
-	(minor) The overall innovation is reasonable, but not super strong. It is mainly composedly of existing techniques: adversarial training + AugMix; and BN + IN. To be fair, packing them together is still well-motivated and smart.


**Time Spent Reviewing:**

5

---

> ### Author Response · Authors · 2021-08-10
> **Response to reviewer iAMx**
>
> Q: Why is AdvMix worse than AugMax?
>
> A: To achieve good model robustness, the augmented training data should contain both enough diversity and enough adversarially hard cases. We discussed this aspect in lines 47-50 in the manuscript.
>
> In AugMax, we optimize the worst-case mixing parameters (w,m) by gradient-based methods while selecting the augmentation parameters randomly (e.g. random rotation angles, random translation pixel numbers). As shown in Figure 1, this leads to a good unification between sample diversity and hard corner-case generation.
>
> In AdvMix, due to the difficulty in directly searching for worst-case augmentation parameters (e.g., translation pixel number, rotation angles, etc.) with gradient-based methods [52], we use the “worst-of-k” method to do adversarial attacks on those augmentation parameters. (See Appendix C for details.) It is less effective to generate adversarially hard examples by searching the augmentation parameters (AdvMix) than the mixing parameters (AugMax). Through the same visualization method as Figure 1, we empirically observe that AdvMix generates less hard-cases than AugMax. We will add the visualization of AdvMix in the revision.
>
> Q: DuBIN boosts AugMix and AugMax with similar margins compared to DuBN, setting doubts on the motivation (feature heterogeneity in AugMax) of adopting DuBIN.
>
> A: We verify the feature heterogeneity by feature visualization: in Figure 1, we show that AugMax samples span a larger area in the feature space and cover more hard-cases compared with AugMix. This motivates us to adopt DuBIN in AugMax.
>
> Our experimental results further justify the need for adapting DuBIN in AugMax. From the results in Section 4.3 of the paper, we actually observe that DuBIN boosts AugMax by larger margins than AugMix. Specifically, as shown in Table 8, on CIFAR10-C, DuBIN boosts AugMax and AugMix by 1.07% and 0.63% respectively compared with DuBN. Similarly, on CIFAR100-C, DuBIN boosts AugMax and AugMix by 1.25% and 0.95% respectively compared with DuBN.
>
>
>
> Q: Training overhead of AugMax?
>
> A:  As shown in lines 69-72, we adopt an efficient adversarial training strategy to increase training efficiency of AugMax. As a result, AugMax adds only a mild amount of extra complexity compared to non-adversarial training methods. For example, AugMax training time is only ~1.5 times that of AugMix on ImageNet. Moreover, AugMax is significantly more efficient than traditional adversarial training which often leads to ~10 times training time overhead compared with AugMix. Please check Table 7 for more details.

---

### Official Review · Reviewer_eujA · 2021-07-17

**Rating:** 6
**Confidence:** 4

**Summary:**

This paper proposes to fuse multiple data augmentations in an adversarial manner. It achieves good robustness compared with AugMix on various natural distortions.

**Limitations And Societal Impact:**

None.

**Main Review:**

Pos:
1. The idea is novel and effective.
2. The paper is well written and easy to understand.

Neg:
1. I would like to see whether the AugMax could be combined with other data augmentations like CutMix.
2. The performance on SA seems to be comparable with AugMix. Could the authors provide results on more data augmentations?

**Time Spent Reviewing:**

1

---

> ### Author Response · Authors · 2021-08-10
> **Response to reviewer eujA**
>
> Q: Can AugMax be combined with other augmentations such as CutMix?
>
> A: Yes, AugMax can be combined with other augmentations such as CutMix. The results on CIFAR10 and CIFAR100 with WRN40-2 backbone are shown below (RA: Robust Accuracy; SA: Standard Accuracy).
>
> CIFAR10:
>
> Normal: SA=94.78%, RA=73.71%
>
> CutMix: SA=96.60%, RA=84.78%
>
> AugMax+CutMix: SA=96.63%, RA=90.69%
>
> CIFAR100:
>
> Normal: SA=76.19%, RA=46.80%
>
> CutMix: SA=79.20%, RA=60.07%
>
> AugMax+CutMix: SA=79.31%, RA=66.37%
>
> As we can see, AugMax+CutMix slightly improves SA and largely improves RA compared with CutMix.
>
>
>
> Q: Results of other augmentations.
>
> A: While AugMax can be considered a data augmentation strategy, its primary goal is to improve model robustness against natural corruptions, rather than increase standard accuracy (SA) on clean images. We therefore focus our comparisons of AugMax with the previous state-of-the-art defense against natural corruptions, AugMix. Our results show that AugMax achieves new state-of-the-art performance against natural corruptions.
>
> To properly address your remark, however, we also run additional experiments and provide the results on CutMix (see our response to the previous question) and show that AugMax outperforms CutMix in terms of natural corruption robustness.

---

### Author Response · Authors · 2021-08-10
**General Response**

We would like to thank the reviewers for their thoughtful reviews and are glad to see that all evaluations are positive. We offer point-by-point answers to each reviewer’s comments below.

---

### Decision · Program_Chairs · 2021-09-27

**Decision:**

Accept (Poster)

**Comment:**

This paper focuses on an interesting new data augmentation called AugMax. The proposal is to reconcile diversity with hardness by learning an adversarially weighted combination of multiple random augmentations. Since this stronger augmentation leads to more diverse input distribution that is harder to fit, they further designed a new normalization strategy called DuBIN to disentangle the instance-wise feature heterogeneity of AugMax samples. The philosophy behind sounds quite interesting to me, namely, diversity-hardness tradeoff of data augmentations. This philosophy leads to a novel algorithm design I have never seen.

The clarity and novelty are clearly above the bar of NeurIPS. While the reviewers had some concerns on the significance, the authors did a particularly good job in their rebuttal. Thus, all of us have agreed to accept this paper for publication! Please include the additional experimental results in the next version.